# Comparative Analysis of the Potential Adaptability of Tibetan Dzo and Yellow Cattle Based on Blood Indices, Metabolites, and Fecal Microbiota

**DOI:** 10.3390/ani14182728

**Published:** 2024-09-20

**Authors:** Kenan Li, Guorui Zhang, Mengjiao Sun, Maolin Xia, Ruizhi Shi, Yanmei Jin, Xiaoqing Zhang

**Affiliations:** 1Grassland Research Institute of Chinese Academy of Agricultural Sciences, Hohhot 010010, China; likenan@caas.cn (K.L.); 17806260775@163.com (G.Z.); 2College of Prataculture, Qingdao Agricultural University, Qingdao 266200, China; 17806274680@163.com; 3Tibet Autonomous Region Animal Husbandry Station, Lhasa 850000, China; 4Institute of Practaculture Science, Tibet Academy of Agricultural and Animal Husbandry Sciences, Lhasa 850000, China; r231119569@163.com; 5Marine College, Shandong University, Weihai 264209, China

**Keywords:** Tibet, dzomo, yellow cattle, blood metabolite profiles, fecal microbiota

## Abstract

**Simple Summary:**

Tibetan yellow cattle and dzo are important meat sources for people living on the Tibetan Plateau. As a hybrid offspring of Tibetan yellow cattle and yaks, dzo inherit the advantages of their parent generations. However, the differences in the blood metabolites and ecological adaptability of Tibetan yellow cattle and dzo remain unclear. In this study, we have explored the potential difference in environmental adaptability between these two breeds of animals by analyzing their blood-based physiological and biochemical parameters, serum metabolites, and fecal microorganisms. The results revealed that dzo showed better adaptation to the high-altitude and low-temperature environment of the Tibetan Plateau, as compared to Tibetan yellow cattle. This study provides a basis for understanding the adaptive mechanism of dzo in the Qinghai–Tibet Plateau.

**Abstract:**

This study aimed to investigate the differences in environmental adaptability between dzo and Tibetan yellow cattle by using corresponding assay kits to analyze blood indices, utilizing mass spectrometry for blood metabolite profiling, and performing 16S rDNA sequencing of fecal microbiota. Forty female cattle were randomly divided into a dzomo (female dzo) group (MG, n = 20) and a Tibetan-yellow-cattle group (HG, n = 20). After 150 days of uniform feeding, six cattle from each group were randomly picked for jugular blood sampling and collection of fecal microorganisms. The results showed that the serum albumin, creatinine, total protein, superoxide dismutase, IgG, and IgM concentrations in the MG group were higher (*p* < 0.05), whereas the serum triglyceride concentration was lower, compared to the HG group (*p* < 0.05). The higher level of phospholipids containing long-chain polyunsaturated fatty acids (PUFAs) (PC (18:5e/2:0), PC (20:5e/2:0), LPC 18:2, LPC 20:5) observed in the serum of the dzo suggests that they have an advantage in adapting to the challenging conditions of the plateau environment. The fecal microbiota analysis showed that *Akkermansia* was significantly enriched in the MG group; this might be the key bacterial genus leading to the strong adaptability of dzo. Our findings indicated the dzo’s superior adaptation to the Tibetan Plateau’s harsh environment.

## 1. Introduction

The Qinghai–Tibet Plateau (extremely cold, with low oxygen levels and strong ultraviolet rays), known as the ‘roof of the world’, covers a region of 2,500,000 km^2^, with altitudes ranging from 3000 to 5000 m. This plateau is rich in animal breeds, and these breeds play an important role in the livestock production systems of China. Yaks are grazing livestock that live in the special geographical environment of the Qinghai–Tibet Plateau all year round. The harsh natural environment of the plateau and the imbalance in the supply of grassland forage between seasons have enabled yaks to develop unique physiological characteristics and extremely strong fiber-degradation capabilities in the process of co-evolution with nature, so as to resist adverse conditions such as alpine cold, low oxygen, high radiation, long withered-grass seasons, and geographical areas characterized by nutrient deficiency [1]. Tibetan yellow cattle are an ancient breed with a long history in the local region; it can live in the meadow grassland at an average altitude of 2300–3800 m [2]. However, the limitations of the breed, such as their diminutive stature, sluggish growth, subpar reproductive capabilities, and inadequate milk production, have hindered their ability to meet the requirements for the sustainable advancement of the Tibetan livestock industry. Dzo, the hybrid offspring of male yak and Tibetan female yellow cattle, are endemic to the Tibetan Plateau and possesses high economic value. (Dzomo are the female of the species.) Dzo males are sterile, but they can gain by heterosis. Their growth performance and meat production are superior to those of their parental generation [3]. Many studies have demonstrated that different animal breeds possess varying abilities to adapt to their environments, based on analyses of their blood’s physio-biochemical features and metabolites. For example, Zhao et al. [4] reported that the Dazu black goat, Nanjiang brown goat, and Saanen dairy goat have different adaptabilities to low-altitude environments, owing to their different red blood cell counts, hemoglobin levels, total protein levels, and albumin concentrations. Wang et al. [5] compared the blood-based physiological and biochemical indices of Qianhua Merino mutton sheep, Blackhead Suffolk sheep, and small-tailed Han sheep, and found that Blackhead Suffolk sheep had better environmental adaptability and disease resistance. Cui et al. [6] found that the crossbreeding progeny of the crossbreed between small-tailed Han sheep and Australian White sheep had better environmental adaptability than small-tailed Han sheep. However, the differences in the blood metabolites and ecological adaptability between yellow cattle and dzo remain unknown. 

Previous studies have found that some microbial features of the rumen are heritable and could be influenced by host genetics, and that different species and breeds of ruminants may have their own stable and heritable microbiota, possibly as a result of co-evolution and adaptation to the host [7]. Some studies have found that the rumena of dzo contain microorganisms that have the ability to degrade lignin and cellulose, enabling the animals to adapt to the plateau environment [1,8]. In addition to rumen microorganisms, it has also been found that the intestinal microorganisms attached to the intestinal tract also play an important role in maintaining the health of the host [9,10]. Therefore, some researchers use feces to study the intestinal microorganisms of ruminants [11,12]. The feed conversion rate, meat quality, and milk quality of yellow cattle, dzo, and yaks grown in the same area are not the same. In addition to genetic differences, we speculate that microorganisms also play an important role in causing these differences. Different parts of the intestine have unique microenvironments, and maintaining such microenvironments is crucial for the health of the entire intestinal system [13]. Furthermore, previous studies have reported that a large number of isolated lymphoid nodules and muscle sarcomeres exist in the caecum of yaks, which enables yaks to have stronger mucosal immunity and store and ferment more food, so that they can better adapt to the special environment of the Tibetan Plateau [14]. Although a large amount of evidence has been obtained from studies on the structure and immune capabilities of different parts of the yak intestine, indicating that it can better adapt to the special environment of the Tibetan Plateau [15], there are few reports on the special intestinal flora formed by dzo under the stress of harsh environments such as low levels of oxygen, cold, shortage of forage supply, and high ultraviolet radiation; this is the second genome of animals, developed to help them adapt to these special environments. 

Currently, there are few reports about the adaptability of Tibetan yellow cattle and dzo to the Tibetan Plateau environment. This study aimed to explore the differences in environmental adaptation between two groups, dzomo and Tibetan yellow cattle, in a plateau environment, by considering three aspects: blood indices, blood metabolites, and fecal microbiota. The study aims to elucidate the physiological response characteristics and microbiological traits that underpin their respective forms of environmental adaptability. The study hypothesizes that dzo, due to their hybrid nature, exhibit superior adaptability to the harsh conditions of the Tibetan Plateau compared to Tibetan yellow cattle. This study provides a basis for understanding the adaptive mechanism of cattle–yak in Qinghai–Tibet Plateau.

## 2. Materials and Methods

### 2.1. Animals and Management

The experiment was conducted from March to July 2022 at the Yak Economic Hybridization Qushui Experimental Station in the Tibet Autonomous Region. The experimental procedure followed the ethical standards of the Welfare and Ethics Committee of the Chinese Association for Laboratory Animal Sciences (SAC/TC 281). In total, 20 female adult dzomo (female yellow cattle × male yak, approximately 7–8 years of age; with an average bodyweight of 161 ± 18.20 kg) and 20 female adult yellow cattle (approximately 7–8 years of age; with an average bodyweight of 155 ± 15.70 kg) were provided by the Yak Economic Hybridization Qushui Experimental Station. The cattle were randomly divided into two breeding groups (n = 20 each): the dzomo group (MG) and the Tibetan-yellow-cattle group (HG). All of the experimental cattle were fed identical diets of concentrate, oat hay, and corn silage on the Tibetan Plateau, using an automatic TMR mixer wagon. All animals were raised for 150 days. Notably, the ratio of concentrate to silage in the mixed feed was 3:7, and nutrient levels for the diet are shown in Table 1. The composites of the diet were measured according to the methods published by the Association of Official Analytical Chemists (AOAC) [16], including those for dry matter (DM, AOAC method 930.15), crude protein (CP, AOAC method 984.13), ether extract (EE, AOAC method 920.39), and crude ash (Ash, AOAC method 924.05). Neutral detergent fiber and acid detergent fiber were analyzed using the ANKOM fiber analysis equipment (A2000i; Ankom Technology Corp., Fairport, NY, USA), as described by Van Soest et al. [17].

### 2.2. Blood Sample and Fecal Sample Preparation

Blood samples were collected on the last day of the experiment, prior to the morning feed. Six cattle with similar body weights were randomly selected from each group. For one set of tests, 2–3 mL of whole blood was collected from the jugular vein of each animal, using vacuum blood collection tubes containing anticoagulant, for the detection of blood-based physiological indicators. For another set of tests, blood was collected, using vacuum blood collection tubes without anticoagulant, in order to separate serum for the analysis of biochemical parameters and antioxidant and immune status, as well as for metabolomics profiling. The whole-blood samples were stored at –20 °C, while the serum samples were preserved at –80 °C for subsequent analysis. On the last day of the feeding trial, to ensure the sterility of samples, sterile disposable gloves were worn as fresh fecal samples were collected from the rectums of experimental animals; the samples were then stored at −80 °C for subsequent 16S rDNA analysis.

### 2.3. Analyses of Blood Samples

The physiological parameters of the whole-blood samples, including the white blood cell (WBC) count, neutrophil (Neu) count, lymphocyte (Lym) count, monocyte (Mon) count, and eosinophils (Eos) count, as well as the corresponding percentages (Neu%, Lym%, Mon%, Eos%), in addition to the red blood cell count (RBC), HGB, hematocrit (HCT), mean corpuscular volume (MCV), mean corpuscular hemoglobin (MCH), mean corpuscular hemoglobin concentration (MCHC), red cell distribution width—coefficient of variation (RDW-CV), red cell distribution width—standard deviation (RDW-SD), platelet count (PLT), mean platelet volume (MPV), platelet distribution width (PDW), and platelet hematocrit (PCT), were determined using the corresponding assay kits purchased from Shenzhen Myriad Biomedical Electronics Co. (Shenzhen, China). The analyses were performed using an automatic blood cell analyzer (BC-5000 Vet, Shenzhen Mindray Animal Medical Technology Co., Ltd., Shenzhen, China) according to the instructions of the kit manufacturer.

The biochemical parameters of the serum samples, including albumin (ALB), urea, creatinine (CREA), triglyceride (TG), and total protein (TP) concentrations, were determined using the corresponding assay kits purchased from Shenzhen Myriad Biomedical Electronics Co. (Shenzhen, China). The analyses were performed using an automatic biochemistry analyzer (BS-240 VET, Shenzhen Mindray Animal Medical Technology Co., Ltd.) according to the instructions of the kit’s manufacturer.

The antioxidant and immune parameters of the serum samples after centrifugation, including the superoxide dismutase (SOD), glutathione peroxidase (GSH-Px), total antioxidant capacity (T-AOC), malondialdehyde (MDA), immunoglobulin G (lgG), and immunoglobulin M (lgM), were determined using the corresponding assay kits purchased from Shenzhen Myriad Biomedical Co., Ltd. (Shenzhen, China).

### 2.4. Metabolomics Analysis

The serum samples were thawed at 4 °C before analysis. For the non-targeted metabolomics assay, 100 μL of serum was placed in an EP tube, and 400 μL of 80% (*v*/*v*) methanol in water was added and vortexed. The extraction mixture was then stored at −20 °C for 2 h. After centrifugation at 20,000× *g* for 10 min, the supernatants were transferred into new 1.5 mL EP tube and dried. The dried extract was stored at −80 °C. The dried extract was then reconstituted in 100 μL of precooled 80% methanol, and an equal part of each sample was taken as a polled quality control (QC) sample.

Samples were acquired by LC-MS, in order, ensuring cross-sorting. Chromatographic separations used a Thermo Scientific UltiMate 3000 HPLC (Waltham, MA, USA) with an ACQUITY UPLC T3 column (100 mm B7 × 2.1 mm, 1.8 µm, Waters, UK) at 50 °C and 0.3 mL/min flow. The mobile phase consisted of phase A (water, 0.1% formic acid) and phase B (acetonitrile, 0.1% formic acid). Gradient elution conditions were set as follows: 0~0.8 min, 2% B; 0.8~2.8 min, 2% to 70% B; 2.8~5.6 min, 70% to 90% B; 5.6~8 min, 90% to 100% B; 8~8.1 min, 100% to 2% B; and 8.1~10 min, 2% B. Each 4 µL sample was injected, starting with WASH and QCs, then QC every 10 samples, and ending with 2 QCs.

A Q-Exactive mass spectrometer (Thermo Scientific) was employed to gather first- and second-order spectra of metabolites eluted from a column in both positive and negative ion modes. Precursor spectra (70–1050 *m*/*z*) were acquired at 70,000 resolution, while fragments were captured at 17,500 resolution. AGC targets were 3 × 10^6^ and 1 × 10^5^, respectively. The DDA mode prioritized the top 3 ions, with maximum injection times of 100 ms and 50 ms. Fragmentation energies ranged from 20 to 60 eV. ESI parameter settings were as follows: spray voltage (|KV|)) was 4000 (positive ion mode) and 4000 (negative ion mode), sheath gas flow rate was 35, aux gas flow rate was 10, and capillary temperature was 320 °C.

Raw mass spectrum data was imported into Compound Discoverer 3.1.0 (Thermo Fisher Scientific, Waltham, MA, USA) for pretreatment: peak extraction, retention time (RT) correction, adduct ion merging, gap filling, background labeling, and metabolite ID. RT and *m*/*z* were matched as to ion ID. Peak intensities were recorded, and molecular weights, RTs, peak areas, and IDs were exported. Metabolites were annotated using the Kyoto Encyclopedia of Genes and Genomes (KEGG) database (https://www.genome.jp/kegg/pathway.html, accessed on 11 April 2023) and the Human Metabolome Database (HMDB) (https://hmdb.ca/metabolites, accessed on 11 April 2023) by matching exact masses, names, and formulas (≤10 ppm difference). MetaX was used to preprocess peak intensities; it removed features <50% in QC or 80% in biosamples, and imputed missing peaks using the k-nearest neighbor algorithm. Principal components analysis (PCA) detected outliers and evaluated batch effects. Probabilistic quotient normalization (PQN) normalized the data. QC-based LOESS corrected signal drift. Metabolic features with coefficients of variation (CV) > 30% across the QC samples were removed. *T*-tests detected metabolite differences between phenotypes, which were subsequently adjusted with FDR (Benjamini–Hochberg). Partial least squares discriminant analysis (PLS-DA) via metaX identified variables, using a threshold of VIP > 1 for importance. KEGG enriched significantly different metabolites (ratio ≥ 2 or ≤1/2, *p* ≤ 0.05, VIP > 1).

### 2.5. Fecal Microbiome Analysis

The total DNA of the microorganisms in the fecal samples was determined using the CTAB according to the manufacturer’s instructions. The total DNA was eluted in 50 μL of elution buffer and stored at −80 °C until measurement in the PCR device acquired from LC-Bio Technology Co., Ltd., Hang Zhou, Zhejiang Province, China. The V3-V4 region of the bacterial 16S rRNA was amplified by PCR. The bacterial primers used in the current study were 341F: 5′-CCTACGGGNGGCWGCAG-3′ and 805R: 5′-GACTACHVGGGTATCTAATCC-3′. The 5′ ends of the primers were tagged with specific barcodes for each sample and for sequencing universal primers. PCR amplification was carried out in a total volume of 25 μL of reaction mixture, comprising 25 ng of template DNA, 12.5 μL of PCR premix, 2.5 μL of each primer, and PCR-grade water to adjust the final volume. The PCR conditions for amplifying the prokaryotic 16S rRNA fragments included an initial denaturation step at 98 °C for 30 s, followed by 32 cycles of denaturation at 98 °C for 10 s, an annealing at 54 °C for 30 s, and an extension at 72 °C for 45 s. This was concluded with a final extension step, at 72 °C for 10 min. Using 2% agarose gel electrophoresis, PCR products were verified. Ultrapure water served as negative control during DNA extraction to prevent false positives. PCR products were purified with AMPure XT beads and quantified by Qubit. Amplicon pools ready for sequencing were evaluated on an Agilent 2100 Bioanalyzer (Agilent Technologies, Santa Clara, CA, USA) for size and with Illumina’s Library Quantification Kit (Illumina, San Diego, CA, USA) for quantity. Sequencing was performed on the NovaSeq PE 250 platform.

Samples were sequenced on an Illumina NovaSeq platform, following LC-Bio’s manufacturer’s recommendations. Paired-end reads were assigned to samples by barcode, truncated, merged with FLASH, and quality filtered under specific conditions using fqtrim (v0.94) to obtain high-quality clean tags. Chimeric sequences were filtered with Vsearch (v2.3.4). After DADA2 dereplication, we obtained feature table and sequences. Alpha and beta diversity were calculated by normalizing sequence counts. Feature abundance was then normalized relative to each sample using SILVA 138. Alpha diversity, analyzed through Chao1, Shannon, and Simpson indices, quantified the species diversity complexity within the samples; this was calculated using QIIME2. Beta diversity was also calculated by QIIME2 and graphed with R. Sequence alignment was performed with BLAST, annotating feature sequences with SILVA. Other diagrams were created with R (v3.5.2).

### 2.6. Statistical Analysis

The independent sample *T*-test of blood-based physiological and biochemical indicators and antioxidant and immune indicators was conducted using SPSS (version 27.0) software. Statistical significance was set to *p* < 0.05. The results are presented as the mean ± standard deviation.

## 3. Results

### 3.1. Blood-Based Physiological and Biochemical Indices in the MG and HG Groups

As shown in Figure 1, the serum ALB (Figure 1A), CREA (Figure 1B), and TP (Figure 1C) concentrations of the MG group were significantly higher (*p* < 0.05) than those of the HG group. In contrast, the TG (Figure 1D) concentration of the MG group was significantly lower (*p* < 0.05) than that of the HG group. There were no significant between-group differences in the other parameters (Table 2 and Table 3).

### 3.2. Antioxidant and Immune Indicators in the MG and HG Groups

As shown in Figure 2, the SOD (Figure 2C) concentration in the MG group was significantly higher (*p* < 0.05) than that in the HG group, while there were no significant differences in the other parameters between the two groups.

As shown in Figure 3, the lgG and lgM concentrations of the MG group were significantly higher (*p* < 0.05) than those of the HG group.

### 3.3. Blood Metabolite Profiling of the MG and HG Groups

The PLS-DA analysis plot showed that the serum metabolites in the MG group were clearly separated from the HG group (Figure 4A,B). The first principal component (PC1) in Figure 4A (R^2^ = 0.97, Q^2^ = 0.51) and the first principal component (PC1) in Figure 4B (R^2^ = 0.96, Q^2^ = 0.73) both met the requirements, with a model prediction rate Q^2^ > 0.5 and explanation rate R^2^ close to 1. This indicates that both of these models were stable and reliable.

A total of 122 differential metabolites were identified between the two groups, with 65 in cation mode and 57 in anion mode. Among these, 71 were upregulated, while 51 were downregulated (Figure 5A,B).

### 3.4. Identification of Differential Metabolites and Metabolic Pathways

As shown in Table 4, a total of 37 differential metabolites were identified in the two groups. Of the metabolites selected from the MG and HG groups, 31 metabolites were related to lipid metabolism, of which 22 metabolites were upregulated and nine metabolites were downregulated. The metabolites were involved in nine metabolic pathways, including the cholinergic synaptic, glycerophospholipid metabolism, insulin secretion, neuroactive ligand–receptor interaction, linoleic acid metabolism, and steroid hormone biosynthesis metabolic pathways.

Six metabolites related to amino acid metabolism were identified, with three being upregulated and three being downregulated. These metabolites were involved in six different metabolic pathways, specifically, the tryptophan metabolism pathway; the arginine and proline metabolism pathway; and the glycine, serine, and threonine metabolism pathway, as well as the lysine degradation metabolism pathway.

We conducted KEGG pathway enrichment analysis. Among all analyzed, the top 20 functional pathways enriched by KEGG for differential metabolites in the cation mode and anion mode between the MG group and HG group are shown in Figure 6. Under the cation mode, the differential metabolites between the MG group and HG group are mainly enriched in arginine and proline metabolism. Under the negative mode, the differential metabolites between MG group and HG group are mainly enriched in purine metabolism, vitamin digestion and absorption, and the HIF-1 signaling pathway.

### 3.5. Fecal Microbiome Profiling of the MG and HG Groups

#### 3.5.1. Number of OTUs

As shown in the Venn diagram in Figure 7A, there were 5937 OTUs of fecal microbiome in the MG group and 6717 in the HG group. Of these OTUs, 2347 were found in both groups.

#### 3.5.2. Alpha Diversity Analysis

The microbiota diversity, as measured by the Shannon index, Simpson index, and Chao 1 index, is shown in Figure 7B. The results showed that there were no significant differences in the Chao1, Shannon, and Simpson indices between the two groups (*p* > 0.05).

#### 3.5.3. PCoA Analysis

The PCoA analysis plot showed that the fecal microorganism in the MG group were clearly separated from the HG group (Figure 7C).

#### 3.5.4. Analysis of the Phylum and Genus Level

In terms of bacterial phylum (Figure 8A), Firmicutes was the dominant phylum in the samples (70.00% in the MG group and 76.94% in the HG group). The relative abundance levels of Bacteroidota in the MG and HG groups were 11.86% and 10.09%, respectively. Campylobacterota and Deinococcota were only found in the HG group. Fibrobacterota was only found in the MG group.

In terms of bacterial genus (Figure 8B), *UCG_005* was the dominant genus in the samples (16.31% in the MG group and 19.10% in the HG group). The relative abundance of *Akkermansia* in the MG group was 12.56%, while the HG group had only 1.95%. 

#### 3.5.5. Differences in Fecal Bacterial Composition Were Analyzed between the MG and HG Groups

The LDA Effect Size (LEfSe) analysis tool was used to identify specialized microbial communities in the MG and HG groups. Species with LDA scores greater than the set value (set to 3.0) are shown, and the length of the histogram represents the magnitude of the effect of the differential species (Figure 8C). The results of LEfSe analysis showed that *g__UCG_005*, *g_Eisenbergiella*, *g_Lachnospiraceae_NK3A20_group*, *g_Raoultibacter*, *g_Lachnospiraceae_XPB1014_group*, *g_Enterococcus*, *g_Catenisphaera*, *g_Vibrio*, *g_Agathobacter*, *g_Erysipelatoclostridium*, *g_Saccharimonadales_unclassified*, *g_Mogibacterium*, *g_Gastranaerophilales_unclassified*, *g_Wolbachia*, *g_Dietzia*, and *g__Solobacterium* in the MG group were significantly lower than in the HG group. The *g__p_2534_18B5_gut_group_unclassified* and *g__Akkermansia* in the MG group were significantly higher than in the HG group.

## 4. Discussion

### 4.1. The Differences in the Blood-Based Physiological and Biochemical Characteristics of the MG and HG Groups

Blood physiology and biochemical characteristics are essential indicators of an animal’s health and nutritional status. For example, TG is a crucial intermediate metabolite involved in lipid metabolism [18], and increased serum TG indicates that fat synthesis is weakened, or catabolism is enhanced [19]. In this study, compared to the animals in the HG group, the serum TG concentrations of the animals in the MG group were significantly lower. Additionally, the muscle fat content of cattle–yak was measured at 1.80%, which was significantly higher than that of Tibetan yellow cattle, which was 1.27% [20,21]. These results indicate that the MG-group animals possess a stronger ability of fat synthesis and a weaker lipolysis ability, as compared to the HG-group animals. The enhanced ability to synthesize fat contributes to both the improved quality of dzo beef and the breed’s heightened ability of adaptation to the alpine environment of the Tibetan Plateau. Serum protein plays a vital role in the stability of blood osmotic pressure, transport of nutrients, supply of body proteins, and maintenance of the animal’s immune performance [22]. The TP, ALB, globulin, and SOD concentrations can serve as important indicators of the oxidative stress and cellular immune function of animals [23,24]. Increased serum TP and ALB levels imply an increased protein-utilization rate. In this experiment, significantly greater serum TP and ALB concentrations were found in the MG-group animals compared to the HG-group animals, indicating that the animals in MG group have stronger digestion abilities and a more capable absorption function for protein. Additionally, the higher lgG and lgM concentrations in the MG group indicated that the MG-group animals had stronger immunity than the HG-group animals [25]. The above results indicate that, compared with Tibetan yellow cattle, the dzo has stronger fat synthesis, protein metabolism, and immune function abilities, thereby improving its adaptability to the Qinghai–Tibet Plateau environment of low temperatures and nutritionally deficient geography. However, it is crucial to note that the conclusions of this study are primarily based on observations under specific conditions and may be influenced by various factors, such as geographical location, seasonal variations, body weight differences, and feeding-management practices. Consequently, to gain a more comprehensive understanding of the adaptive differences between dzo and Tibetan yellow cattle under varying environmental conditions, future research should replicate experiments and validations across different geographical regions and seasons, and under various feeding-management conditions. Such a research design will contribute to a more accurate assessment of the universality of the dzo’s adaptability and its underlying mechanisms, thereby providing a more scientific basis for the development of animal husbandry in the Qinghai–Tibet Plateau region.

### 4.2. The Differences in the Blood Metabolite Profiles of the MG and HG Groups

#### 4.2.1. The Differences in the Lipid Metabolism Levels of the MG and HG Groups

Metabolomics is able to reveal the relationship between genetic information and physiological phenotypes in animals. Glycerophospholipids are phosphatidyl compounds composed of phosphatidic acid and hydroxyl-containing compounds, including phosphatidylcholine (PC), phosphatidylethanolamine (PE), lysophosphatidylcholine (LPC), and lipid-phosphatidylethanolamine (LPE). They are essential components of biological membranes and lipid metabolism, participate in cellular energy metabolism and lipid uptake, and influence intramuscular fat deposition through phospholipid metabolism. In this study, seven LPCs, five PCs, four LPEs, one sphingomyelin (SM), and one lyso-phosphatidic acid (LPA) were upregulated, while three PCs, one LPC, one LPE, and one LPG were downregulated in the MG group, compared to the HG group. These results indicate that the MG animals had significantly different phospholipid metabolism levels, compared to the HG animals. The detailed comparison shows that the upregulated metabolites in the comparison between the MG group and the HG group, including PC (18:5e/2:0), PC (20:5e/2:0), LPC 18:2, and LPC 20:5, are all phospholipids containing long-chain polyunsaturated fatty acids (PUFAs). The primary constituent of cell membranes is phospholipids, and long-chain PUFAs exhibit greater fluidity under low-temperature conditions. Therefore, the increases in the content levels of phospholipids containing long-chain PUFAs may be related to the stronger plateau-adaptability of dzo compared to Tibetan yellow cattle. A recent study showed that the higher content of phospholipids containing long-chain PUFAs in dzo muscle, compared to cattle, may be one of the reasons for the dzo’s adaptation to the plateau environment [26]. In addition, studies have demonstrated that linoleic acid (an important polyunsaturated fatty acid) can reduce blood cholesterol and TG levels, enhance lipid and cholesterol efflux, and promote the release of anti-inflammatory factors [27,28]. In this experiment, the upregulation of linoleic acid in the MG group probably led to a significantly lower serum TG concentration in this group of animals (see Figure 1). Acetylcholine and its precursor choline act as cell-signalling molecules to mediate anti-inflammatory responses [29]. In this experiment, compared to the HG group, the MG group exhibited downregulated choline and upregulated linoleic acid and acetylcholine enriched in the glycerophospholipid metabolic pathway. This indicates that the MG-group animals had better anti-inflammatory abilities.

Prostaglandin G2 can be converted into an inflammatory mediator, E2, and its content can be downregulated to inhibit platelet activity and reduce inflammation in the body [30]. Yao et al. [31] reported that dehydroepiandrosterone can modulate some signalling pathways and attenuate oxidative stress and inflammation triggered by oleic acid stimulation in laying hens. Cao et al. [32] reported that dehydroepiandrosterone increased the total antioxidant capacity and superoxide dismutase activity, but decreased the level of reactive oxygen species in lipopolysaccharides (LPS)-induced mice. Jiao et al. [33] reported that tetrahydrocorticosterone was significantly upregulated in yak calves supplemented with concentrate, giving them stronger immunity. Therefore, the significantly downregulated prostaglandin G2 and trans-leukotriene C4, as well as the considerably upregulated dehydroepiandrosterone and tetrahydrocorticosterone, in the MG group of this study indicate that the MG-group animals have stronger anti-inflammatory and antioxidant abilities compared to the HG-group animals. This was also confirmed by the higher serum superoxide dismutase activity in the MG group.

The levels of phospholipid metabolites containing long-chain PUFAs, such as PC (18:5e/2:0), PC (20:5e/2:0), LPC 18:2, and LPC 20:5, were higher in the serum of dzo compared to Tibetan yellow cattle, indicating that dzo have an advantage in adapting to the plateau environment.

#### 4.2.2. The Differences in Amino Acid Metabolism and Nucleotide Metabolites in the MG and HG Groups

Amino acids are essential components of living organisms and play crucial roles in protein digestion, as well as antioxidant, anti-inflammatory, and immune regulation [34]. Taylor et al. [35] reported that lysine can promote the expression of lipolysis genes and the synthesis of glycogen in the skeletal muscles of mice and inhibit the process of intracellular fat synthesis in muscle cells. In the current experiment, compared to the HG group, the MG group exhibited a downregulation of (N6, N6, N6)-trimethyl-l-lysine, which is metabolized from the lysine metabolic pathway. This is likely attributed to the fat-synthesis ability of dzo, which is also a reason for the lower serum TG concentration of the dzo. This is in line with the results of Chen et al. [36], who observed reduced fat synthesis efficiency in the livers of mice and increased serum TG. In addition, the significantly upregulated CREA in the MG group, enriched in the arginine and proline metabolic pathways, led to a significantly higher serum concentration, compared to the HG group (Figure 1B). Since the blood CREA level is closely related to the muscle activity of animals [37], the upregulation of CREA indicates the excellent potential meat-producing performance of the MG-group animals. Studies have demonstrated that the blood metabolites tryptophan, xanthurenic acid, and indole-3-lactic acid have anti-inflammatory and antioxidant properties [27,38]. Previous studies have reported that administering indole to enteropathy mice reduced inflammation induced by the innate immune response and restored intestinal dysbiosis [39], while xanthurenic acid promoted vascular smooth muscle activity and neuro-immunity in mice [40]. Similar to the lipid metabolites, significantly upregulated xanthurenic acid and indole-3-lactic acid in the MG group reflect higher anti-inflammatory and antioxidant or immune activity in the MG-group cattle.

#### 4.2.3. KEGG Enrichment Analysis of Differential Metabolites

To gain a deeper understanding of the physiological processes and functions involved in the differential metabolites between the MG group and the HG group, we performed KEGG annotation and enrichment analysis on these differential metabolites. Under the cation mode, the differential metabolites between MG group and HG group are mainly enriched in arginine and proline metabolism. The differential metabolites enriched in this pathway include CREA, which we have discussed earlier. Arginine and proline participate in RNA synthesis and protein glycosylation, which are the premise and guarantee required for cells to function [41]. Proline is not only the main amino acid that maintains cell structure and function, but also an important regulator of cell metabolism and normal physiological functions. It plays a role in protein synthesis, metabolism, and immune and antioxidant reactions [42]. Under the negative mode, the differential metabolites between MG group and HG group are mainly enriched in purine metabolism, vitamin digestion and absorption, and the HIF-1 signaling pathway. Vitamins are trace organic substances necessary for maintaining normal growth and development of the organism. They cannot be synthesized in animals, and play a very important role in the growth, development, and enhancement of the disease resistance of animals [43]. The enrichment of vitamin digestion and absorption pathways contributes to the adaptability of dzo to the harsh environment of the Qinghai–Tibet Plateau. Similar to our findings, Lv et al. [44] found that the enhanced absorption of vitamin A and the upregulation of arginine and proline pathways enabled grazing Tibetan sheep to better adapt to the cold environment on the Qinghai–Tibet Plateau. HIF-1α can form different signal pathways with various upstream and downstream proteins to mediate hypoxia signals, regulate a series of compensatory reactions to hypoxia, and play an important role in the growth and development and the physiological and pathological processes of the body [45]. It is suggested that the dzo can adapt to the hypoxic environment of the Qinghai–Tibet Plateau through the HIF-1 signaling pathway. Purine metabolism refers to the in vivo synthesis and decomposition of nucleic acids, bases, adenine, guanine, and other purine derivatives in the body. Studies have shown that xanthine and its derivatives can produce a large amount of reactive oxygen species in the process of metabolizing into uric acid, causing oxidative stress damage to the vascular endothelium [46]. However, further in-depth research is needed to understand the effects of purine metabolism on the vascular endothelium of dzo. In summary, the differential metabolites in the blood of dzo and Tibetan yellow cattle are primarily enriched in arginine and proline metabolism, and vitamin digestion and absorption, as well as the HIF-1 signaling pathway, thereby playing significant roles in protein metabolism, physiological regulation, and immune responses. Given the limitations of our experimental conditions and the number of experimental animals, we would recommend conducting studies which were more extensive and in-depth, as well as under different environmental conditions, to validate and expand upon the findings of this research. These studies could include evaluations of the adaptability of dzo and cattle across various geographical regions, climate types, and ecosystems, and under different management practices.

### 4.3. The Differences in the Fecal Microbiome Profiles of the MG and HG Groups

Many studies have already underscored the significance of gastrointestinal-tract bacteria, which are intimately linked to the health and production performance of economically important animals [47]. Therefore, studying the gastrointestinal microbiota and its potential functions can contribute to enhancing the production performance of livestock on the Qinghai–Tibet Plateau, as well as to gaining insights into the microbiological mechanisms that enable local livestock to adapt to the harsh conditions of the plateau. According to the alpha diversity data, there was no significant difference in fecal microbial diversity between the MG and HG groups, which may be caused by the close kinship between the two breeds. Most studies suggest that Firmicutes and Bacteroidetes are the dominant phyla in the human gut, as well as in those of other mammals [48], which is consistent with the findings of this study. Furthermore, the results showed that the relative abundance of *Akkermansia* in the MG group was significantly higher than that in the HG group. *Akkermansia* can utilize host mucin as a carbon source when food is insufficient [49]. In the course of the year, yaks mainly obtain nutrients by grazing on natural pastures in the Qinghai–Tibet Plateau, and in the long withered-grass season, yaks can only obtain nutrients by grazing on withered grass. Therefore, the increase in the abundance of *Akkermansia* in the intestine may indicate that yaks are under severe nutritional stress during the withered-grass season; this abundance provides an effective buffer for changes in nutrient intake, enabling the host to adapt to the seasonal fluctuations in food supply and thus adapt to the plateau environment. Similar to our results, a recent study also found that *Akkermansia* is an important bacterial genus in the intestine of yaks [50]. The *Akkermansia* genus helps the organism improve the energy-conversion efficiency under cold stress to resist the threat from the cold [51]. This may be a positive response of the organism’s microorganisms, enabling the animalst to improve energy utilization efficiency when the yaks are subjected to cold stress in winter and spring. However, the function of *p_2534_18B5_gut_group_unclassified* is still unclear. In addition, the *UCG_005*, *Eisenbergiella*, *Lachnospiraceae_NK3A20_group*, *Raoultibacter*, *Lachnospiraceae_XPB1014_group*, *Enterococcus*, *Catenisphaera*, *Vibrio*, *Agathobacter*, *Erysipelatoclostridium*, *Saccharimonadales_unclassified*, *Mogibacterium*, *Gastranaerophilales_unclassified*, *Wolbachia*, *Dietzia*, and *Solobacterium* in the MG group were significantly lower than in the HG group. The fibrolytic bacteria, *Lachnospiraceae* (*Lachnospiraceae_NK3A20_group* and *Lachnospiraceae_XPB1014_group*), and *Ruminococcaceae* (*UCG_005*) are critical for gastrointestinal tract fermentation [52]. These results suggest that these intestinal microorganisms play an important role in improving the fiber-digestion function of Tibetan yellow cattle. *Erysipelatoclostridium* is a strict anaerobe, but in the latest bacterial classification manual, *Erysipelatoclostridium* is an aerobic and facultative anaerobic bacteria, so its specific functions need further study [53]. Due to the nature of the intestinal microbiota, many functions of unclassified bacteria may be underestimated. In order to further understand the effect of bacterial group increase on intestinal function, functional analysis of intestinal microbiota should be carried out in combination with metagenomics and/or metabolomics analysis.

## 5. Conclusions

Based on the blood-based physiological parameters, biochemical parameters, blood metabolites, and fecal microbiota, dzo demonstrate better adaptation to the high-altitude and low-temperature environment of the Tibetan Plateau, as compared to Tibetan yellow cattle. The level of phospholipids containing long-chain PUFAs, such as PC (18:5e/2:0), PC (20:5e/2:0), LPC 18:2, and LPC 20:5 in blood, as well as the abundance of *Akkermansia* in feces, contribute to improving the dzo’s adaptability to the harsh environment of the Qinghai–Tibet Plateau. These findings provide information and references for understanding the dzo’s adaptation to the plateau environment. Future research also needs to combine the metagenomics, metabolomics, and transcriptomics to explore the adaptive mechanisms of dzo relative to the Qinghai–Tibet Plateau environment from multiple angles.

## Figures and Tables

**Figure 1 animals-14-02728-f001:**
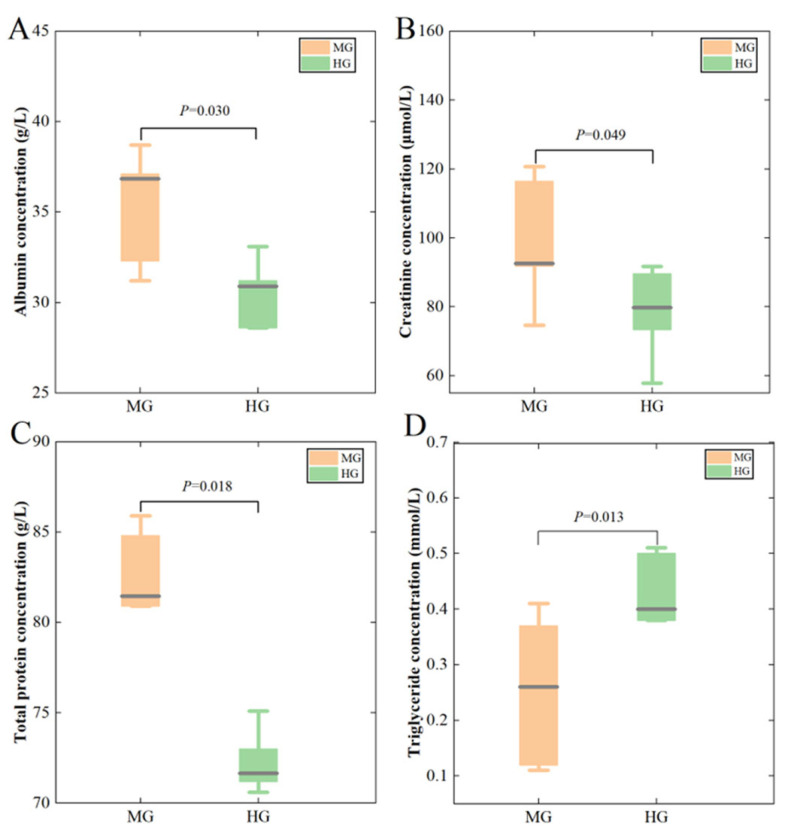
Differences in serum biochemical indices in the MG and HG groups. Albumin concentration (**A**), Creatinine concentration (**B**), Total protein concentration (**C**), and Triglyceride concentration (**D**).

**Figure 2 animals-14-02728-f002:**
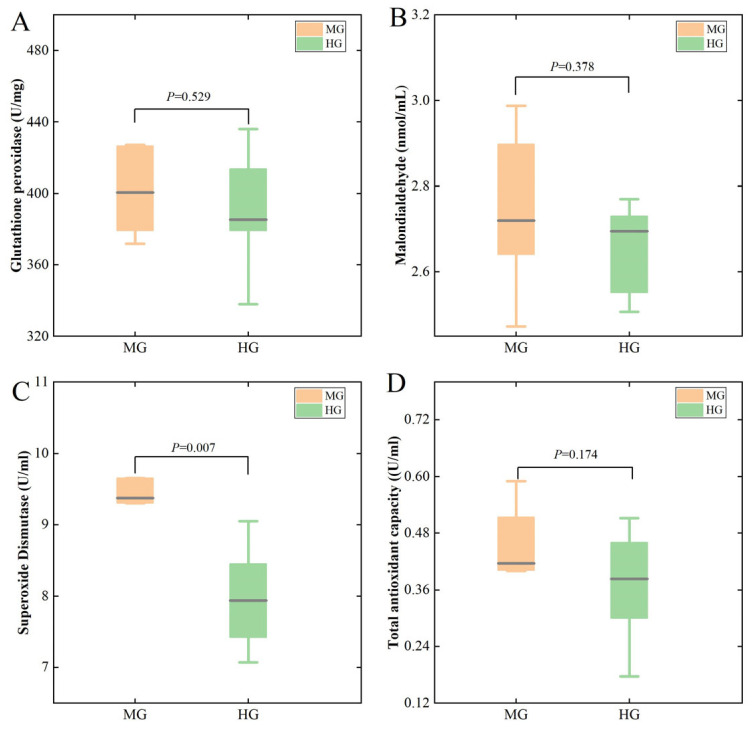
Differences in serum antioxidant indices in the MG and HG groups. Glutathione peroxidase (**A**), Malondialdehyde (**B**), Superoxide dismutase (**C**), and Total antioxidant capacity (**D**).

**Figure 3 animals-14-02728-f003:**
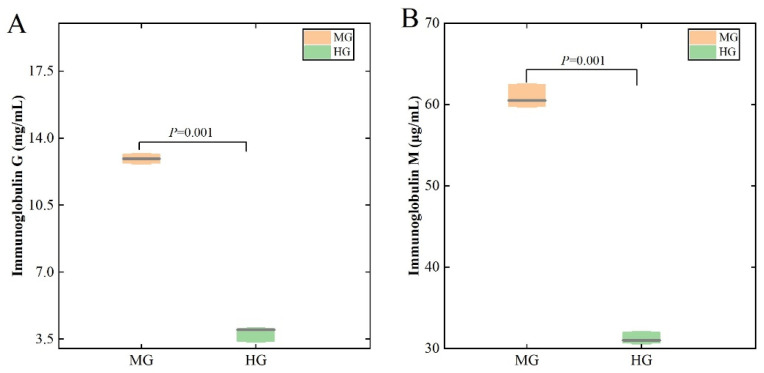
Differences in serum immune indices in the MG and HG groups. Immunoglobulin G (**A**) and Immunoglobulin M (**B**).

**Figure 4 animals-14-02728-f004:**
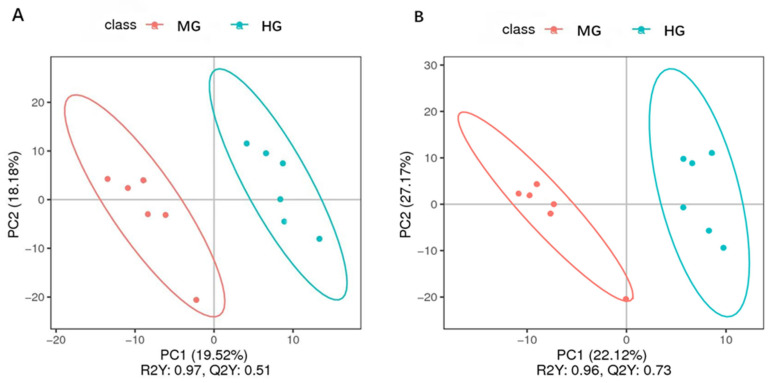
PLS-DA analysis plot in positive model (**A**) and negative model (**B**) in the MG and HG groups.

**Figure 5 animals-14-02728-f005:**
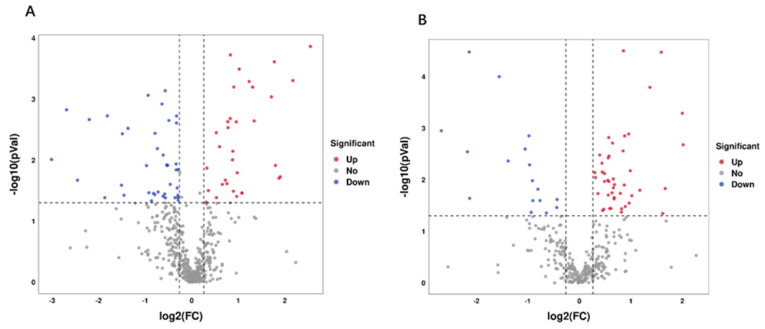
Volcano plots of the metabolites in the positive model (**A**) and negative model (**B**) in the MG and HG groups. In the differential analysis, red points indicate upregulated metabolites, whereas blue points correspond to downregulated ones. The *X*-axis is the mean ratio fold-change of the relative abundance of each metabolite, and the *Y*-axis represents the statistical significance or *p*-value of the ratio fold-change for each metabolite.

**Figure 6 animals-14-02728-f006:**
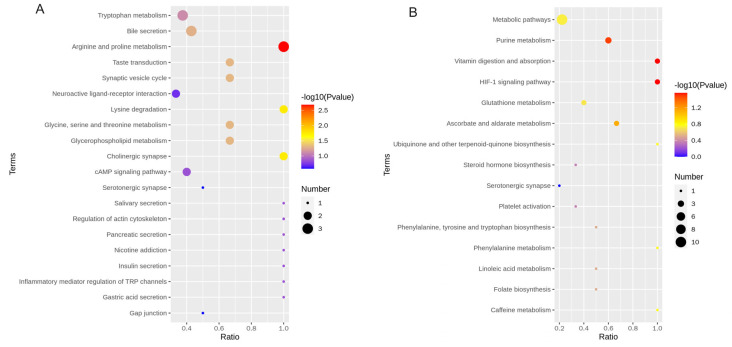
Functional enrichment analysis of differential metabolites. The KEGG pathway enrichment diagrams for the MG group and HG group in positive ion mode (**A**) and negative ion mode (**B**), respectively.

**Figure 7 animals-14-02728-f007:**
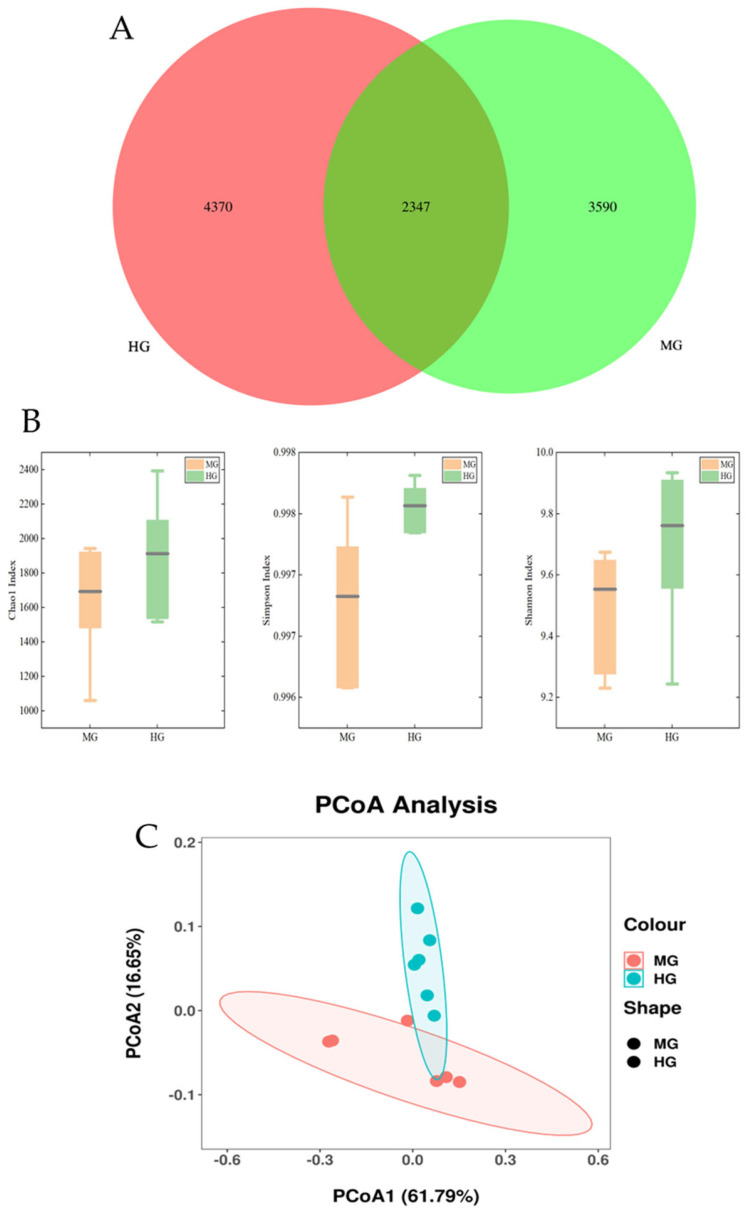
Differences in alpha and beta diversity of fecal microbes between the MG and HG groups: Venn diagram of fecal microbiota in the MG and HG groups (**A**); the α−diversity of the Shannon index, Simpson index, and Chao 1 index of fecal microbes (**B**); and PCoA analysis plot in the MG and HG groups (**C**).

**Figure 8 animals-14-02728-f008:**
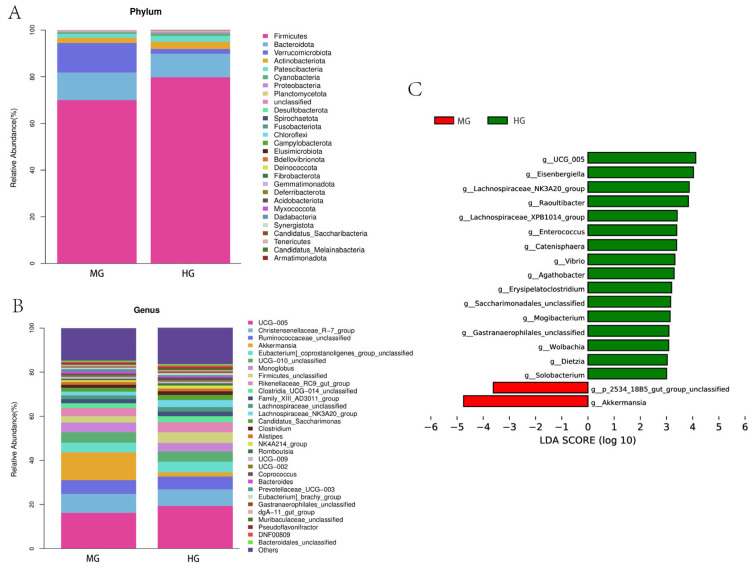
The composition of and differences in fecal microbes between the MG and HG groups: the relative abundance of bacterial compositions at the phylum level in the MG and HG groups (**A**); the relative abundance of bacterial compositions at the genus level in the MG and HG groups (**B**); and the differences in bacteria composition in the MG and HG groups (genus-level) (**C**).

**Table 1 animals-14-02728-t001:** Chemical composition of the diet (DM basis, %).

Item	Diet
Ingredients	
Corn	14.60
Wheat bran	3.70
Soybean meal	3.60
Rapeseed meal	1.95
Wheat	3.70
CaHPO_4_	0.15
NaCl	0.30
Premix ^1^	2.00
Oat hay	25.00
Corn silage	45.00
Total	100.00
Chemical composition	
CP	10.91
EE	1.68
ADF	35.2
NDF	54.26
P	0.16
Ca	0.83
Ash	9.51
Metabolic energy (MJ/Kg) ^2^	8.96

NDF: neutral detergent fiber; ADF: acid detergent fiber; CP: crude protein; EE: ether extract. ^1^ The premix provided the following per kg of diet: Cu 10 mg; Fe 65 mg; Mn 30 mg; Zn 25 mg; I 0.5 mg; Se 0.1 mg; Co 0.1 mg; VA 4000 IU; VD 500 IU; and VE 40 IU. ^2^ Nutrient levels were all determined by measurement, except for the metabolic energy. The calculation method employed for metabolic energy refers to standard guidance as to nutrient requirements and foodstuff compositions of beef cattle.

**Table 2 animals-14-02728-t002:** Serum physiological indices of Dzo and Tibetan yellow cattle.

Item	Dzo	Yellow Cattle	*p*-Value
WBC (10^9^/L)	10.04 ± 3.91	9.06 ± 1.82	0.589
Neu (10^9^/L)	3.66 ± 1.95	3.24 ± 0.94	0.645
Lym (10^9^/L)	5.73 ± 1.64	5.13 ± 0.81	0.436
Mon (10^9^/L)	0.15 ± 0.14	0.11 ± 0.04	0.509
Eos (10^9^/L)	0.50 ± 0.50	0.59 ± 0.38	0.748
Neu (%)	35.02 ± 11.81	35.40 ± 6.45	0.946
Lym (%)	59.50 ± 12.30	57.27 ± 6.95	0.707
Mon (%)	1.30 ± 0.70	1.18 ± 0.29	0.714
Eos (%)	4.18 ± 2.66	6.15 ± 3.35	0.287
RBC (10^12^/L)	8.18 ± 1.46	8.03 ± 1.39	0.858
HGB (g/L)	142.33 ± 20.61	141.33 ± 12.72	0.921
HCT (%)	42.73 ± 7.01	41.63 ± 2.99	0.731
MCV (fL)	52.48 ± 3.91	52.65 ± 5.84	0.955
MCH (pg)	17.52 ± 1.29	17.85 ± 1.73	0.713
MCHC (g/L)	334.17 ± 7.08	339.50 ± 8.60	0.268
RDW-CV (%)	21.50 ± 1.86	21.13 ± 1.30	0.700
RDW-SD (fL)	41.30 ± 5.68	40.30 ± 3.52	0.722
PLT (10^9^/L)	311.00 ± 126.85	317.67 ± 67.24	0.912
MPV (fL)	5.97 ± 0.43	6.25 ± 0.64	0.389
PDW (%)	15.78 ± 0.40	15.63 ± 0.27	0.467
PCT (%)	0.18 ± 0.07	0.20 ± 0.04	0.642

**Table 3 animals-14-02728-t003:** Serum biochemical indices of Dzo and Tibetan yellow cattle.

Item	Dzo	Yellow Cattle	*p*-Value
AST (U/L)	55.52 ± 11.96	64.12 ± 18.98	0.370
ALT (U/L)	24.28 ± 8.53	27.67 ± 12.40	0.594
Glu-G (mmol/L)	3.07 ± 1.16	3.66 ± 0.54	0.291
TBA (μmol/L)	16.40 ± 13.89	11.95 ± 3.14	0.462
γ-GT (U/L)	18.05 ± 4.37	23.63 ± 14.57	0.390
UREA (mmol/L)	4.55 ± 0.29	4.88 ± 0.77	0.348
T-Bil-V (mol/L)	2.16 ± 1.09	1.80 ± 0.38	0.470
TC (mmol/L)	2.23 ± 0.35	2.96 ± 0.88	0.091
LDL-C (mmol/L)	0.34 ± 0.08	0.41 ± 0.31	0.069
HDL-C (mmol/L)	1.34 ± 0.19	1.40 ± 0.24	0.624
UA (μmol/L)	30.93 ± 6.48	38.60 ± 9.36	0.130
Ca (mmol/L)	1.97 ± 0.29	1.80 ± 0.13	0.232
P (mmol/L)	1.97 ± 0.55	2.36 ± 0.78	0.340
Mg (mmol/L)	0.99 ± 0.10	1.01 ± 0.06	0.707

**Table 4 animals-14-02728-t004:** Identification and change trends of the differential metabolites in the MG and HG groups.

Chemical Denomination	Formula	*p*-Value	VIP	m. z	Retention Time (min)	Trend	Metabolic Pathway
Lipid metabolism-related metabolites							
PC (16:1e/6:0)	C_30_H_60_NO_7_P	2.00 × 10^−4^	1.57	578.42	11.38	up	
PC (14:1e/2:0)	C_24_H_48_NO_7_P	1.60 × 10^−2^	1.84	494.32	8.67	up	
PC (18:5e/2:0)	C_28_H_48_NO_7_P	3.20 × 10^−2^	1.57	542.32	8.46	up	
PC (16:2e/2:0)	C_26_H_50_NO_7_P	4.10 × 10^−2^	1.45	520.34	9.19	up	
PC (20:5e/2:0)	C_30_H_52_NO_7_P	4.80 × 10^−2^	1.51	570.35	9.17	up	
PC (22:3e/12:0)	C_42_H_80_NO_7_P	4.10 × 10^−2^	1.08	742.57	10.54	down	
PC (18:3e/2:0)	C_28_H_52_NO_7_P	3.70 × 10^−2^	1.71	546.35	9.20	down	
PC (18:1/18:3)	C_44_H_80_NO_8_P	3.40 × 10^−2^	1.54	826.56	11.78	down	
SM (d14:1/16:2)	C_35_H_67_N_2_O_6_P	7.00 × 10^−3^	1.18	643.48	11.73	up	
LPC 15:1	C_23_H_46_NO_7_P	3.40 × 10^−2^	1.74	480.31	8.66	up	
LPC 14:1	C_22_H_44_NO_7_P	2.80 × 10^−2^	1.47	510.28	8.01	up	
LPC 16:1	C_24_H_48_NO_7_P	4.20 × 10^−2^	1.53	538.32	8.87	up	
LPC 18:2	C_26_H_50_NO_7_P	7.00 × 10^−3^	1.48	564.33	9.17	up	
LPC 16:0	C_24_H_50_NO_7_P	9.00 × 10^−3^	1.79	540.33	9.51	up	
LPC 20:5	C_28_H_48_NO_7_P	1.20 × 10^−2^	1.55	586.32	8.63	up	
LPC 20:1	C_28_H_56_NO_7_P	2.30 × 10^−2^	1.72	594.38	10.32	up	
LPC 20:3	C_28_H_52_NO_7_P	4.20 × 10^−2^	1.40	590.35	9.48	down	
LPE 16:0	C_21_H_44_NO_7_P	3.00 × 10^−3^	2.17	452.28	9.45	up	
LPE 17:0	C_22_H_46_NO_7_P	1.00 × 10^−2^	1.60	466.29	9.74	up	
LPE 16:1	C_21_H_42_NO_7_P	3.90 × 10^−2^	1.25	450.26	8.83	up	
LPE 18:1	C_23_H_46_NO_7_P	3.20 × 10^−2^	1.62	478.29	8.87	up	
LPE 20:3	C_25_H_46_NO_7_P	1.00 × 10^−3^	1.87	502.29	9.45	down	
LPA 16:0	C_19_H_39_O_7_P	4.00 × 10^−3^	1.47	409.24	9.51	up	
LPG 18:1	C_24_H_47_O_9_P	4.10 × 10^−2^	1.30	511.30	6.58	down	
Choline	C_5_H_13_NO	1.40 × 10^−2^	1.44	104.11	1.28	down	Cholinergic synapse Glycerophospholipid metabolism
Acetylcholine	C_7_H_15_NO_2_	6.00 × 10^−3^	1.74	146.12	1.37	up	Cholinergic synapse Glycerophospholipid metabolism Insulin secretion cAMP signaling pathway Neuroactive ligand–receptor interaction
Linoleic acid	C_18_H_32_O_2_	3.00 × 10^−3^	1.44	279.23	9.17	up	Linoleic acid metabolism
Tetrahydrocorticosterone	C_21_H_34_O_4_	6.00 × 10^−3^	1.40	349.24	7.97	up	Steroid hormone biosynthesis
11-transleukotriene C4	C_30_H_47_N_3_O_9_S	2.20 × 10^−2^	1.44	624.30	5.02	down	
Dehydroepiandrosterone (DHEA)	C_19_H_28_O_2_	6.00 × 10^−4^	1.89	271.21	8.42	up	
Prostaglandin G2	C_20_H_32_O_6_	2.50 × 10^−2^	1.69	367.21	7.85	down	Platelet activation
Amino acids metabolism-related metabolites							
Xanthurenic acid	C_10_H_7_NO_4_	5.00 × 10^−4^	2.29	206.04	5.30	up	Tryptophan metabolism
Indole-3-lactic acid	C_11_H_11_NO_3_	3.00 × 10^−3^	1.06	206.08	5.65	up	
Serotonin	C_10_H_12_N_2_O	1.20 × 10^−2^	1.14	177.10	1.60	down	Tryptophan metabolism Inflammatory mediator regulation of TRP channels cAMP signaling pathway
Creatinine	C_4_H_7_N_3_O	1.30 × 10^−2^	1.20	114.07	1.32	up	Arginine and proline metabolism
Creatine	C_4_H_9_N_3_O_2_	2.40 × 10^−2^	1.90	132.08	1.37	down	Arginine and proline metabolism Glycine, serine and threonine metabolism
N6,N6,N6-Trimethyl-l-lysine	C_9_H_20_N_2_O_2_	4.10 × 10^−2^	1.36	189.16	1.25	down	Lysine degradation

PC: phosphatidylcholine; SM: sphingomyelin; LPC: lysophosphatidylcholine; LPE: lipid-phosphatidylethanolamine; LPA: lyso-phosphatidic acid; LPG: lysophosphatidyl glycerol.

## Data Availability

Dataset available on request from the corresponding author.

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
