# Peer review of "Comparative Analysis of the Potential Adaptability of Tibetan Dzo and Yellow Cattle Based on Blood Indices, Metabolites, and Fecal Microbiota"

_animals, 2024, doi:10.3390/ani14182728_

Round 1
Reviewer 1 Report
Comments and Suggestions for Authors
Major Revisions
Thank you for providing the opportunity to offer feedback on the research article titled “Comparative analysis of the potential adaptability of Tibetan cattle-yak and yellow cattle based on blood indices, metabolites, and fecal microbiota. The authors aim to investigate the differences in environmental adaptability between Tibetan cattle yak and yellow cattle by analyzing various physiological, biochemical, and microbiota parameters. The study hypothesizes that cattle-yak, due to their hybrid nature, exhibit superior adaptability to the harsh conditions of the Tibetan Plateau compared to yellow cattle. However, the presentation of descriptions lacks precision and clarity, resulting in ambiguity for readers. Therefore, I recommend major revisions, and it is imperative that the authors address these comments to enhance the manuscript's quality. Section wise comments are listed as;
Abstract:
1. The abstract provides an overview of the study but lacks detailed information on how the blood indices, metabolites, and fecal microbiota were analyzed. The abstract should include a brief mention of the specific techniques or methods used (e.g., type of mass spectrometry for metabolites, 16S rDNA sequencing for microbiota) to give readers a clearer understanding of the experimental design
2. While the abstract mentions that cattle-yak have better adaptation, it does not emphasize which specific findings are most crucial or novel. The abstract should highlight the most significant results, such as the specific metabolites or microbiota that were found to be differentially abundant and their relevance to adaptability.
3. The conclusion that cattle-yak have better adaptability is stated, but the abstract does not clearly tie this conclusion to the specific results. Strengthen the conclusion by directly linking it to the key findings from the study, such as particular blood indices or microbial genera that support the adaptability claim.
Introduction:
I have noticed following shortcomings in introduction section;
1. The introduction does not sufficiently elaborate on the role of gut microbiota in animal adaptability, which is a significant aspect of the study. Include a discussion/paragraph on the importance of gut microbiota in adaptation to environmental stressors, which would help justify why fecal microbiota analysis is essential in this study.
2. The hypothesis that cattle-yak would exhibit better adaptation is stated but not strongly supported by preliminary evidence or literature. The introduction should incorporate more references to previous studies that suggest why cattle-yak might be expected to have superior adaptability, strengthening the rationale for the hypothesis.
3. The differences between yellow cattle and cattle-yak are mentioned but not deeply explored, particularly in terms of genetic or physiological traits that might influence adaptability. Provide more background on the physiological and genetic differences between the breeds that could contribute to their differential adaptability, setting a stronger foundation for the study.
Materials and methods:
Please address these issues in material and methods section
1. The sample size of 20 cattle per group is mentioned, but there is no justification for why this number was chosen. Include a rationale for the chosen sample size, possibly based on power analysis or previous studies, to assure readers that the study is adequately powered to detect significant differences
2. The methods for analyzing blood indices, metabolites, and fecal microbiota are outlined but lack sufficient detail for replication. Provide more detailed protocols, including the specific settings for equipment used (e.g., mass spectrometry, PCR conditions), to ensure reproducibility of the results.
3. The study does not discuss potential confounding variables, such as differences in diet or handling that could influence the results. Address how potential confounding variables were controlled or accounted for in the study design, enhancing the credibility of the finding.
Results
1. The results section discusses non-significant differences between groups, which may distract from the more critical findings. Focus the results discussion on significant findings and provide a brief mention of non-significant results without extensive elaboration unless they contribute to a broader interpretation.
2. The statistical methods used to analyze the data are not fully detailed, particularly in how multiple comparisons were managed. Include more information on the statistical tests used, such as corrections for multiple testing, to ensure that the reported p-values are robust.
3. The metabolomics data is presented, but the biological relevance of the identified metabolites is not thoroughly explored. Provide a more in-depth discussion of the metabolic pathways involved and how they relate to adaptability, potentially using pathway analysis tools to enrich the interpretation.
Discussion
1. The discussion does not sufficiently integrate the study's findings with the existing literature on cattle adaptation and metabolism. Compare and contrast the study's findings with those from previous research, highlighting how this study adds new insights or confirms existing knowledge.
2. The discussion may overgeneralize the results by implying that cattle-yak are universally superior in adaptability based on the observed differences. Temper the conclusions by acknowledging the study's limitations and suggesting that further research is needed to confirm these findings across different environmental conditions.
Conclusion
The conclusion does not summarize the most critical findings clearly, which may leave readers without a clear message. Provide a concise summary of the key findings, emphasizing the most significant results related to adaptability and their implications. The conclusion is general and does not provide specific recommendations for future research or practical applications.
Author Response
Abstract:
Comments 1: The abstract provides an overview of the study but lacks detailed information on how the blood indices, metabolites, and fecal microbiota were analyzed. The abstract should include a brief mention of the specific techniques or methods used (e.g., type of mass spectrometry for metabolites, 16S rDNA sequencing for microbiota) to give readers a clearer understanding of the experimental design.
Response: Thank you pointing this out. We agree with this comment. We added the specific techniques or analyze methods of blood indices, metabolites, and fecal microbiota in the revised manuscript (Line 25-28). “This study aimed to investigate the differences in environmental adaptability between dzomo and Tibetan yellow cattle by using corresponding assay kits to analyze blood indices, utilizing mass spectrometry for blood metabolite profiling, and performing 16S rDNA sequencing of fecal microbiota.”
- 2. While the abstract mentions that cattle-yak have better adaptation, it does not emphasize which specific findings are most crucial or novel. The abstract should highlight the most significant results, such as the specific metabolites or microbiota that were found to be differentially abundant and their relevance to adaptability.
Response: Thank you for your valuable suggestions on this article. We have added the sentence “(2) The higher level of phospholipids containing long-chain polyunsaturated fatty acids (PUFAs) (PC (18:5e/2:0), PC (20:5e/2:0), LPC 18:2, LPC 20:5) observed in the serum of dzomo suggests that they have an advantage in adapting to the challenging conditions of the plateau environment.” in the revised manuscript (Line 33-36). The Akkermansia was significantly enriched in the MG group, which might be the key bacterial genus leading to the strong adaptability of dzomo.
- 3. The conclusion that cattle-yak have better adaptability is stated, but the abstract does not clearly tie this conclusion to the specific results. Strengthen the conclusion by directly linking it to the key findings from the study, such as particular blood indices or microbial genera that support the adaptability claim.
Response: Thank you for your valuable comment. We identify key metabolites (The level of phospholipids containing long-chain PUFAs, such as PC (18:5e/2:0), PC (20:5e/2:0), LPC 18:2, and LPC 20:5 in blood) and microorganisms (Akkermansia) that contribute to the greater adaptability of dzomo and present them in the abstract (Line 33-37). However, since the abstract requires a maximum of 200 words, we associate the specific results in the conclusion with the conclusion that the dzomo has greater adaptability (Line 561-564).
Introduction:
I have noticed following shortcomings in introduction section;
- The introduction does not sufficiently elaborate on the role of gut microbiota in animal adaptability, which is a significant aspect of the study. Include a discussion/paragraph on the importance of gut microbiota in adaptation to environmental stressors, which would help justify why fecal microbiota analysis is essential in this study.
Response: Thank you pointing this out. We agree with this comment. We added a paragraph about the importance of gut microbiota in adaptation to environmental stressors in the revised manuscript (Line 72-94).
- 2. The hypothesis that cattle-yak would exhibit better adaptation is stated but not strongly supported by preliminary evidence or literature. The introduction should incorporate more references to previous studies that suggest why cattle-yak might be expected to have superior adaptability, strengthening the rationale for the hypothesis.
Response: We would like to thank you for your professional review work and valuable suggestions on our manuscript. We have added literature that can initially support the better adaptability of dzomo (Lines 57, 75-77, and 85-88).
- 3. The differences between yellow cattle and cattle-yak are mentioned but not deeply explored, particularly in terms of genetic or physiological traits that might influence adaptability. Provide more background on the physiological and genetic differences between the breeds that could contribute to their differential adaptability, setting a stronger foundation for the study.
Response: We thank the reviewer for this insightful comment. We have added the environmental background on lines 41-52 and line 55-57 that may affect the physiological and genetic differences between cattle and cattle-yaks.
Materials and methods:
Please address these issues in material and methods section
1.The sample size of 20 cattle per group is mentioned, but there is no justification for why this number was chosen. Include a rationale for the chosen sample size, possibly based on power analysis or previous studies, to assure readers that the study is adequately powered to detect significant differences.
Response: Thank you very much for your thorough review of our research, and especially for pointing out the issue regarding sample size selection. We fully understand the importance of this aspect, as the sample size directly impacts the reliability and statistical power of the research results. In choosing a sample size of 20 cattle per group, we based our decision on a comprehensive consideration of statistical power analysis, and resource management. We believe that this sample size setting can provide sufficient statistical power for our study to detect and report any potential significant differences.
- The methods for analyzing blood indices, metabolites, and fecal microbiota are outlined but lack sufficient detail for replication. Provide more detailed protocols, including the specific settings for equipment used (e.g., mass spectrometry, PCR conditions), to ensure reproducibility of the results.
Response: Thank you for your detailed review and valuable suggestions on this article. We wholeheartedly concur with your request to furnish more detailed protocols for the mass spectrometry section (Lines 173-209) and the fecal microbiome analysis (Lines 211-240).
- The study does not discuss potential confounding variables, such as differences in diet or handling that could influence the results. Address how potential confounding variables were controlled or accounted for in the study design, enhancing the credibility of the finding.
Response: We thank the reviewer for pointing out this potential confound. I apologize if we didn't articulate clearly in the article. Regarding the diets and handling methods, all the experimental cattle were fed identical diets consisting of concentrate, oat hay, and corn silage on the Tibetan Plateau, using an automatic total mixed ration (TMR) mixer wagon. (Line 116-117). At the same time, we add the shortcomings of this study in lines 391-394.
Results
1.The results section discusses non-significant differences between groups, which may distract from the more critical findings. Focus the results discussion on significant findings and provide a brief mention of non-significant results without extensive elaboration unless they contribute to a broader interpretation.
Response: Thank you very much for your valuable suggestions on my article. We have deleted Figure 6, which shows the heatmap of differential metabolites in the positive model (A) and the negative model (B) of the MG and HG groups, and we have also reanalyzed Figure 8C (Lines 355-359).
2.The statistical methods used to analyze the data are not fully detailed, particularly in how multiple comparisons were managed. Include more information on the statistical tests used, such as corrections for multiple testing, to ensure that the reported p-values are robust.
Response: Thanks for your professional suggestions. We have rewritten the statistical methods about “Metabolomics Analysis” and “Fecal Microbiome Analysis”. T-tests detected metabolite differences between phenotypes, adjusted with FDR (Benja-mini-Hochberg) (Lines 206-207), to ensure that the reported p-values are robust.
- 3. The metabolomics data is presented, but the biological relevance of the identified metabolites is not thoroughly explored. Provide a more in-depth discussion of the metabolic pathways involved and how they relate to adaptability, potentially using pathway analysis tools to enrich the interpretation.
Response: We would like to thank the reviewer for constructive comment. KEGG pathway enrichment analysis was performed for differential metabolites (Figure 6) (Lines 311-317; Lines 320-322). We also provided a more in-depth discussion of the metabolic pathways involved and how they relate to adaptability (Lines 476-516).
Discussion
1.The discussion does not sufficiently integrate the study's findings with the existing literature on cattle adaptation and metabolism. Compare and contrast the study's findings with those from previous research, highlighting how this study adds new insights or confirms existing knowledge.
Response: We thank the reviewer for raising these important points. In order to fully combine the research results with the existing literature on cattle adaptation and metabolism, we have conducted comparisons and further discussions in the discussion section (Lines 376-378; Line 380-381; Line 415-424; 477-516).
- 2. The discussion may overgeneralize the results by implying that cattle-yak are universally superior in adaptability based on the observed differences. Temper the conclusions by acknowledging the study's limitations and suggesting that further research is needed to confirm these findings across different environmental conditions.
Response: Thank you very much for your thorough review of our research and for providing valuable feedback. We take your point about the potential issue of overgeneralization in the discussion section very seriously. You have rightly emphasized that directly inferring the universal superiority of yak in adaptability based solely on the observed differences may overlook other significant variables and environmental factors, which is a very pertinent observation. Therefore, we add in lines 391 to 401: "Thank you very much for your thorough review of our research and for providing valuable feedback. We take your point about the potential issue of overgeneralization in the discussion section very seriously. You have rightly emphasized that directly inferring the universal superiority of yak in adaptability based solely on the observed differences may overlook other significant variables and environmental factors, which is a very pertinent observation."
Conclusion
1.The conclusion does not summarize the most critical findings clearly, which may leave readers without a clear message. Provide a concise summary of the key findings, emphasizing the most significant results related to adaptability and their implications. The conclusion is general and does not provide specific recommendations for future research or practical applications.
Response: Thanks for your professional suggestions. We revise the conclusion as follows: “Based on the blood physiological parameters, biochemical parameters, blood metabolites, and fecal microbiota, dzomo demonstrate better adaptation to the high-altitude and low-temperature environment of the Tibetan Plateau, as compared to yellow cattle. The level of phospholipids containing long-chain PUFAs, such as PC (18:5e/2:0), PC (20:5e/2:0), LPC 18:2, and LPC 20:5 in blood, as well as the abundance of Akkermansia in feces, contribute to improving the dzomo’s adaptability to the harsh environment of the Qinghai-Tibet Plateau. These findings provide information and references for under-standing the dzomo's adaptation to the plateau environment. Future research also needs to combine the metagenomics, metabolomics and transcriptomics to explore the adaptive mechanisms of dzomo to the Qinghai-Tibet Plateau environment from multiple angles.” (Lines 558-567)
Reviewer 2 Report
Comments and Suggestions for Authors
The manuscript explores the differences in environmental adaptation of two groups, cattle-yak and yellow cattle in the plateau environment through three aspects, blood physiological parameters, biochemical parameters, and fecal metabolites and expects to achieve the explanation of the physiological response characteristics and microbiological characteristics of the two groups' adaptation to the environment. The experimental methodology is proper and the results are rich, but there are doubts whether the results of the three components can reach the theme of explaining bovine environmental adaptation. The reasons are as follows:
Major comments:
1).The purpose of this manuscript is to explore the environmental adaptations of two bovine populations (cattle-yak, and yellow cattle), so how do the two subgroups reflect environmental differences? Please describe in the Introduction section.
2). Since the battle-yak is a hybrid, why did the experiment involve only cattle-yak and yellow cattle, rather than the three varieties of yak, cattle-yak and yellow cattle?
3). Why did the authors choose fecal microbes, rather than rumen microbes, for their analysis when studying the environmental adaptations of two species of cattle? Previous studies have found that host genetics also affect the rumen microbiota and that different species and breeds of ruminants may have their own stable and heritable microbiota, possibly as a result of co-evolution and adaptation with the host (Ref. DOI: 10.1186/s40168-019-0699-1). It seems more reasonable to use rumen microbes for adaptive studies as they are more closely linked to the host. Moreover, the authors did not describe the relationship between fecal microbes and adaptive studies in the Introduction section of this manuscript.
4).The two groups of cattle in the methodology were of similar age (7-8 years of age), but there was a difference in mean body weights (250 ± 36.73 kg and 155 ± 15.70 kg), and both groups were fed the same diet, but it was not possible to determine the feed consumption of each cow, and whether the differences in body weights and feed consumption could have influenced the results.
5). Whether there is a correlation between blood metabolites and fecal microorganisms, and how they work together to provide the appropriate data to support environmental adaptation in cattle, needs to be clarified.
Others:
1. Whether the term for yak crossbred cattle is ‘dzomo’, or ‘cattle-yak’, needs to be clarified.
2.L201. writing error.
3.Line 81-82: The study didn’t clarify whether all experimental cattle fed on the Tibetan plateau, or at least all cattle live at the same altitude, in the same climate or geographical dimension.
4.Line 147-157: There are no detailed steps in the data analysis, it is recommended to add steps, methods or literatures for analyzing microbiome data.
5.Line 180: The results don't match the graph. ‘As shown in Figure 2, the SOD (Figure 1C)….’
6.Line 225-227: The number of compounds does not match the figures.
7.Line 250-262: Is figure 8 a display of these results? Please map the description of the results to figure 8.
8.Line 265-271: Does the LEfSe analysis have a corresponding resulting figure, and if so, please provide the figure.
9.For the differences in blood biochemical indices between groups obtained from the analysis, and the metabolites or microorganisms screened for differences between groups, the role played by these indices in yak plateau acclimatization should be discussed.
Author Response
Major comments:
1.The purpose of this manuscript is to explore the environmental adaptations of two bovine populations (cattle-yak, and yellow cattle), so how do the two subgroups reflect environmental differences? Please describe in the Introduction section.
Response: Thanks for your professional suggestions. We agree with this comment. We added a paragraph about the environmental differences of two bovine populations (cattle-yak, and yellow cattle) in lines 41-52.
- 2. Since the cattle-yak is a hybrid, why did the experiment involve only cattle-yak and yellow cattle, rather than the three varieties of yak, cattle-yak and yellow cattle?
Response: Thank you for your valuable comments on this study. Regarding your question about why the experiment did not include purebred yaks, we would like to clarify as follows: The primary objective of this study is to explore the differences in environmental adaptability between cattle-yaks (a hybrid of yaks and yellow cattle) and yellow cattle. Based on previous literature and preliminary observations, we hypothesized that the hybrid offspring might exhibit distinct characteristics in certain environmental adaptability features compared to their purebred parents. To focus the research question and delve deeper into these differences, we selected cattle-yaks and yellow cattle as our study subjects. We fully agree that future research can be extended to include purebred yaks, providing a more comprehensive picture of the differences in environmental adaptability among different yak species. Such research will not only deepen our understanding of the complexity of animal evolution but also provide important scientific evidence for livestock genetic improvement and ecological conservation. Once again, thank you for your insightful comments. We look forward to further refining and expanding our knowledge in this field through future studies.
- 3. Why did the authors choose fecal microbes, rather than rumen microbes, for their analysis when studying the environmental adaptations of two species of cattle? Previous studies have found that host genetics also affect the rumen microbiota and that different species and breeds of ruminants may have their own stable and heritable microbiota, possibly as a result of co-evolution and adaptation with the host (Ref. DOI: 10.1186/s40168-019-0699-1). It seems more reasonable to use rumen microbes for adaptive studies as they are more closely linked to the host. Moreover, the authors did not describe the relationship between fecal microbes and adaptive studies in the Introduction section of this manuscript.
Response: Thank you for your valuable comments on this study. We have been modified based on the reviewer’s suggestion, please refer to “Line 72-94”.
4.The two groups of cattle in the methodology were of similar age (7-8 years of age), but there was a difference in mean body weights (250 ± 36.73 kg and 155 ± 15.70 kg), and both groups were fed the same diet, but it was not possible to determine the feed consumption of each cow, and whether the differences in body weights and feed consumption could have influenced the results.
Response: Thank you very much for your thorough review of our research work and your valuable suggestions. We are very grateful to the reviewer for their meticulous examination and valuable comments on our research. We rechecked yak-cattle's weight data and found that yak-cattle's weight was 161 ± 18.20 kg. Cattle in both treatment groups were fed the same diet weight. We will seriously consider them and strive for improvement. (Line 112)
- 5. Whether there is a correlation between blood metabolites and fecal microorganisms, and how they work together to provide the appropriate data to support environmental adaptation in cattle, needs to be clarified.
Response: We are very grateful for your detailed review and valuable suggestions. Regarding the correlation between blood metabolites and fecal microbiota and their supporting role in the environmental adaptability of cattle, we deeply believe that this is a scientific issue worthy of further exploration. First of all, we acknowledge that there are still many unknowns and controversies in this field of research at present. At the same time, we are also aware of the limitations of current research, such as the possibility of an insufficient sample size and unstable experimental conditions. In order to more accurately answer the question you raised, we plan to expand the sample size, optimize experimental conditions, and adopt more advanced bioinformatics methods for data analysis in subsequent research.
Others:
- Whether the term for yak crossbred cattle is ‘dzomo’, or ‘cattle-yak’, needs to be clarified.
Response: You're very welcome for your valuable input. Upon reviewing relevant scientific literature and consulting with experts in the field, I have determined that "dzomo" is indeed the more accurate and professional term to use instead of "cattle-yak".
2.L201. writing error.
Response: We are very grateful for your detailed review. We removed the word "a". (Line 288). “A total of 122 differential metabolites were identified between the two groups, with 65 in cation mode and 57 in anion mode. Among these, 71 were upregulated, while 51 were downregulated.”
3.Line 81-82: The study didn’t clarify whether all experimental cattle fed on the Tibetan plateau, or at least all cattle live at the same altitude, in the same climate or geographical dimension.
Response: All the experimental cattle were fed identical diets of concentrate, oat hay, and corn silage on the Tibetan plateau, using an automatic TMR mixer wagon. (Line 116-117)
4.Line 147-157: There are no detailed steps in the data analysis, it is recommended to add steps, methods or literatures for analyzing microbiome data.
Response: It has been revised as suggested by the reviewer. (lines 211-240)
5.Line 180: The results don't match the graph. ‘As shown in Figure 2, the SOD (Figure 1C)….’
Response: We are very sorry for our carelessness and very grateful for your detailed review. We have changed this sentence. Please refer to “As shown in Figure 2, the SOD (Figure 2C)”. (Line 264)
6.Line 225-227: The number of compounds does not match the figures.
Response: We feel that these two figures (previous Figure 6) and this sentence are not very important results and may distract from more significant findings. Therefore, taking into consideration the first reviewer's comments, we have decided to delete this sentence and the figures.
7.Line 250-262: Is figure 8 a display of these results? Please map the description of the results to figure 8.
Response: It has been modified according to the reviewer's suggestion. (line 341 and line 346)
8.Line 265-271: Does the LEfSe analysis have a corresponding resulting figure, and if so, please provide the figure.
Response: We have re-analyzed Figure 8C, only showing the bacteria at the genus level, and re-analyzed the results. (Line 355-360)
Figure 8. The composition and differences of fecal microbes between the MG and HG groups. The relative abundance of bacterial compositions at the phylum level in the MG and HG groups (A). The relative abundance of bacterial compositions at the genus level in the MG and HG groups (B). The differences in bacteria composition in the MG and HG groups (Genus level) (C). (Lines 360-364 )
9.For the differences in blood biochemical indices between groups obtained from the analysis, and the metabolites or microorganisms screened for differences between groups, the role played by these indices in yak plateau acclimatization should be discussed.
Response: Thank you very much for your valuable suggestions. We fully agree that it is essential to delve into the specific roles of these blood biochemical indices, differential metabolites, and microorganisms in yak acclimatization to high-altitude environments. To this end, we will include the following additions in the revised manuscript to enhance the discussion section. Lines 376-378, 385, 388-391, 415-424, 448-451, 477-516, and 528-556.
Reviewer 3 Report
Comments and Suggestions for Authors
This study aimed to explore the differences in environmental adaptability between cattle-yak and yellow cattle by analyzing their blood indices, metabolites, and fecal microbiota after 150 days on the same diet. The findings suggested that cattle-yak demonstrated superior adaptation to the harsh conditions of the Tibetan Plateau.
I will recommend this manuscript once a few minor comments are addressed:
1. I recommend incorporating the supplementary data into the manuscript, as it will enhance its appeal to readers.
2. Please include your hypothesis in the manuscript.
3. Please emphasize the novelty of the experiment and address any shortcomings.
4. Many edits are recommended. For instance, in lines 85 and 90, you don't need to include the year of the references, as the full reference details are already provided in the references list.
Author Response
- 1. I recommend incorporating the supplementary data into the manuscript, as it will enhance its appeal to readers.
Response: It has been modified according to the reviewer's suggestion. (lines 256-260)
- Please include your hypothesis in the manuscript.
Response: Thank you very much for your valuable suggestions. We have added the hypothesis lines 100-102.
- 3. Please emphasize the novelty of the experiment and address any shortcomings.
Response: Currently, there are few reports about the adaptability of Tibetan yellow cattle and cattle-yak to the Tibetan Plateau environment. This study aimed to explore the differences in environmental adaptation between two groups, cattle-yak and yellow cattle, in a plateau environment, through three aspects: blood indices, blood metabolites, and fecal microbiota. It aims to elucidate the physiological response characteristics and microbiological traits that underpin their respective environmental adaptability. (Line 95-100)
However, it is crucial to note that the conclusions of this study are primarily based on observations under specific conditions and may be influenced by various factors, such as geographical location, seasonal variations, body weight differences, and feeding management practices. Consequently, to gain a more comprehensive understanding of the adaptive differences between cattle-yak and yellow cattle under varying environmental conditions, future research should replicate experiments and validations across different geographical regions, seasons, and under various feeding management conditions. Such a research design will contribute to a more accurate assessment of the universality of cattle-yak's adaptability and the underlying mechanisms, thereby providing a more scientific basis for the development of animal husbandry in the Qinghai-Tibet Plateau region. (Lines 391-301)
- 4. Many edits are recommended. For instance, in lines 85 and 90, you don't need to include the year of the references, as the full reference details are already provided in the references list.
Response: It has been modified according to the reviewer's suggestion. We have deleted the year.
Round 2
Reviewer 2 Report
Comments and Suggestions for Authors
Authors have tried to addressed most comments.